

# *Helicobacter pylori* virulence factors: relationship between genetic variability and phylogeographic origin

Aura M. Rodriguez[1], Daniel A. Urrea[2] and Carlos F. Prada[1]

[1] Grupo de Investigación de Biología y Ecología de Artrópodos. Facultad de Ciencias, Universidad del Tolima, Ibague, Tolima, Colombia
[2] Laboratorio de Investigaciones en Parasitología Tropical. Facultad de Ciencias, Universidad del Tolima, Ibague, Tolima, Colombia

## ABSTRACT

**Background:** *Helicobacter pylori* is a pathogenic bacteria that colonize the gastrointestinal tract from human stomachs and causes diseases including gastritis, peptic ulcers, gastric lymphoma (MALT), and gastric cancer, with a higher prevalence in developing countries. Its high genetic diversity among strains is caused by a high mutation rate, observing virulence factors (VFs) variations in different geographic lineages. This study aimed to postulate the genetic variability associated with virulence factors present in the *Helicobacter pylori* strains, to identify the relationship of these genes with their phylogeographic origin.
**Methods:** The complete genomes of 135 strains available in NCBI, from different population origins, were analyzed using bioinformatics tools, identifying a high rate; as well as reorganization events in 87 virulence factor genes, divided into seven functional groups, to determine changes in position, number of copies, nucleotide identity and size, contrasting them with their geographical lineage and pathogenic phenotype.
**Results:** Bioinformatics analyses show a high rate of gene annotation errors in VF. Analysis of genetic variability of VFs shown that there is not a direct relationship between the reorganization and geographic lineage. However, regarding the pathogenic phenotype demonstrated in the analysis of many copies, size, and similarity when dividing the strains that possess and not the cag pathogenicity island (cagPAI), having a higher risk of developing gastritis and peptic ulcer was evidenced. Our data has shown that the analysis of the overall genetic variability of all VFs present in each strain of *H. pylori* is key information in understanding its pathogenic behavior.

# INTRODUCTION

*Helicobacter pylori* is a gram-negative bacterium that colonizes the stomach of 50% of the global population (*Burucoa & Axon, 2017*; *Khalifa, Sharaf & Aziz, 2010*) varying according to geographical areas, and it is estimated that the frequency of infection in developed

Corresponding author
Carlos F. Prada, cfpradaq@ut.edu.co

countries ranges between 20–40%, while in developing countries the frequency ranges between 70–90% (*Salih, 2009*); particularly high levels observed in South America, sub-Saharan Africa and the Middle East (*Ben Mansour et al., 2016*; *McDonald et al., 2015*; *Peleteiro et al., 2014*). This high level of prevalence of *H. pylori* has been attributed to the poor socioeconomic status and overcrowded conditions with more than three people in the same room (*Cheng et al., 2009*; *Salih, 2009*). The main mechanism of transmission occurs from person to person, with intrafamilial or close community groups spread and is acquired during childhood and established throughout the life of the host (*Fujimoto et al., 2007*).

Chronic active gastritis is caused by the bacteria in all infected subjects; between 10% and 15% of cases progress within a subset of clinical disease manifests as peptic ulcer or chronic atrophic gastritis; and less than 1% of which develop gastric adenocarcinoma, and <0.1% develop gastric lymphoma of mucosa-associated lymphoid tissue (MALT) (*Burucoa & Axon, 2017*; *Correa & Piazuelo, 2012*; *Torre et al., 2015*). This incidence is considered a consequence of the multifactorial nature of infection, in which disease risk and susceptibility is influenced by complex interaction between a ethnicity of host, *H. pylori* genetic diversity and environmental factors (*Cover, 2016*; *den Hollander et al., 2013*; *Lin & Koskella, 2015*).

Due to clinical relevance, in the last decade, several research groups have focused on the complete sequencing of the *H. pylori* genome, which have resulted in obtaining more than 200 complete genomes of strains reported in the different databases (*Cao et al., 2016*; *Thorell, Lehours & Vale, 2017*). Multi-locus sequence typing (MLST) studies and the STRUCTURE Bayesian population cluster method have shown a notable geographical clustering of *H. pylori* genomes across world regions; where it divides into seven major genetically and geographically distinct *H. pylori* populations ('hp'): hpEurope, hpNEAfrica, hpAfrica1, hpAfrica2, hpAsia2, hpSahul and hpEastAsia; and five subpopulations ('hsp'): hpAfrica1 is divided into two subpopulations (hspWAfrica, hspSAfrica) and hpEastAsia is divided into three subpopulations, hspEAsia, hspMaori and hspAmerind (*Falush et al., 2003*; *Kumar et al., 2015*; *Moodley et al., 2012*).

Due to the genetic heterogeneity present within *H. pylori* genomes, bacterial virulence factors (VF) likely play an important role in determining the outcome of *H. pylori* infection (*Wroblewski, Peek & Wilson, 2010*). From these genomes, comparative analyses of genes, especially VF, have been developed; showing a high genetic variability among the strains (*Delahay, Croxall & Stephens, 2018*; *Mucito-Varela et al., 2020*). In this context, several genetic studies have identified about 87 genes associated with VF in *H. pylori* (*Javed, Skoog & Solnick, 2019*; *Wroblewski, Peek & Wilson, 2010*). However, most studies have focused on specific VFs such as the pathogenicity island genes associated with the development of gastric cancer (*Nejati et al., 2018*; *Yakoob et al., 2017*). For example, *H. pylori* strains are frequently segregated into *cagA*-positive and *cagA*-negative strains (one of the most intensely investigated *H. pylori* genes), depending on the presence or absence of the terminal gene product of the cag island, *cagA* (*Wroblewski, Peek & Wilson, 2010*). Another *H. pylori* locus frequently studied is the *vacA* gene, which encodes the secreted toxin *vacA*, have been associated with increases of the gastric cells

permeability, induces apoptosis and suppresses the immune response, among other effects; it is present in the majority of *H. pylori* strains; however, considerable differences in vacuolating activities are observed between strains (*Basso et al., 2008*; *Wroblewski, Peek & Wilson, 2010*). Likewise, the presence of the *dupA* gene is associated with increased susceptibility to develop peptic ulcer disease (*Alam et al., 2020*). Despite others VFs such as ureases, adhesins, Lewis antigen, immune modulator, flagella genes and plasticity zones have been implicated in the pathogenicity of *H. pyroli* (*Cao et al., 2016*; *Kumar et al., 2015*; *Wroblewski, Peek & Wilson, 2010*); the genetic variability between the different sequenced strains has not been as well explored in comparison to other VFs such as *cag* and *vacA* genes.

In this study, we therefore aimed to clarify two important goals; (1) assess relationship between genetic variability of 87 VFs at various levels among 135 genomes of *H. pylori* strains, and (2) the relationship between the genetic variability and/or reorganizations of these genes with their phylogeographic origin and pathogenic phenotype.

## MATERIALS AND METHODS

### Complete genome sequence collection

We downloaded the sequences and gene annotations of 135 complete genomes sequences of *Helicobacter pylori* strains, which are available at the genome resources database from NCBI (https://www.ncbi.nlm.nih.gov/genome/?term=) by December 20, 2019. Incomplete sequences of strains in scaffold or contigs phase were not taken into account to avoid false negatives in the genetic analyses.

Each of the *H. pylori* strains was classified according to its phylogeographic origin: Hp populations as HpEurope (Europe, Middle East, India and Iran), HpAfrica1 (West Africa and South Africa), HpAfrica2 (South Africa), HpAsia2 (Northern India, Bangladesh, Thailand and Malaysia), HpSahul (Australian Aborigines and Papuans of New Guinea), HpEastAsia (East Asia), and Hsp subpopulations as hspWAfrica (West Africa), hspSAfrica (South Africa), hspMaori (Native Taiwanese, Melanesian and Polynesian), hspAmerind (Native Americans), and hspEAsia (East Asians); according to different authors (*Kumar et al., 2015*; *Sayers et al., 2019*; *Wattam et al., 2017*) and the pathogenic phenotype or clinical origin of each one of the genomes, dividing it into four categories: Gastritis, Peptic Ulcer, gastric lymphoma of mucosa-associated lymphoid tissue (MALT) and Gastric Adenocarcinoma. The pathogenic classification was established in several scientific articles, summarized in the PATRIC (The Pathosystems Resource Integration Center) database (*Wattam et al., 2017*) and NCBI data base (*Sayers et al., 2019*). List of these *H. pylori* strains, provides the GenBank Reference IDs, their phylogeographic origins and pathogenic phenotypes is summarized in the Table S1.

### Identification and genetic annotation confirmation of virulence factors (VF)

A search in the different scientific databases and articles was carried out, taking as reference the list of "Virulence factors database (VFDB)" (*Liu et al., 2019*); selecting 87 genes associated to virulence factors that were most closely related to the pathogenic
phenotype, described for *Helicobacter pylori* (*Liu et al., 2019*; *Sayers et al., 2019*; *Wattam et al., 2017*). The 87 genes associated with virulence factors analyzed in this study were classified into seven groups according to the metabolic function in the bacteria, taking into account databases and literature (*Javed, Skoog & Solnick, 2019*; *Liu et al., 2019*; *Sayers et al., 2019*). This classification is summarized in Table 1.

## Presence and copy number analysis of virulence factors

First, from the gene annotation in each of the 135 genomes, a contingency table was generated identifying the presence, orientation and location in coordinates for each VF by genome. In order to generate this data, nucleotide sequences of the 87 virulence factors from the *H. pylori* (strain ATCC 26695) genome were downloaded using as a reference. A basic local alignment search tool (BLAST) was performed, using the nucleotide and amino acid sequences of each VF gene against each analyzed genomes (E-value less than 0.01 and ≥85% of identity and ≥70% of coverage); to corroborate the presence and position of each gene through a binary matrix of presence (1) and/or absence (0) (coordinate and position matrix, plus/plus or plus/minus). All gene annotations were confirmed using BEACON program (Bacterial Genome Annotation Comparison) (*Kalkatawi, Alam & Bajic, 2015*).

After generating the binary matrix of presence and/or absence (1, 0), duplicated genes (paralogous genes) were complemented in this matrix, using the following methodology: (a) Identification of copies in the gene annotations in the GenBank Flat files by command line, (b) Using the nucleotide and aminoacid sequences of the reference paralogous gene(s) (*e.g., cag* gene, to identify the presence of *cag1* to *cag5*) using blast2seq (https://blast.ncbi.nlm.nih.gov/Blast.cgi), and (c) MUSCLE multiple alignments (*Edgar, 2004*) of homologous regions with possible duplicated genes, using the Geneious platform (*Kearse et al., 2012*). Additional copies for each gene were identified in the matrix as two or more duplicate genes. Based on the data, a cluster analysis was performed using the (hclust) package, which performs a hierarchical cluster analysis using different dissimilarity methods in R (*Charif & Lobry, 2007*). Euclidean distances between strains were identified using ward.D2 method (Ward's minimum variance clustering), generating a dendrogram by UPGMA (*Gronau & Moran, 2007*).

## Position and synteny analysis of virulence factor

Based on the confirmation of the gene annotations, rearrangements present in each *H. pylori* strain (inversions, translocations, deletions, duplications and insertions of large regions in the genome) were identified by a paired comparison of reference strains (strain ATCC 26695) against each VF gene by local alignment with BLAST2seq (https://blast.ncbi.nlm.nih.gov/Blast.cgi). Similarly, Geneious program (*Kearse et al., 2012*) was used to perform MUSCLE multiple alignments (*Edgar, 2004*), extracting the sequences with the exact coordinates of VF location, verifying the presence and position of each gene.

In order to verify both the annotations and the position of the genes in the genomes, we performed a comparison of synteny analyses between VF in each *H. pylori* strains, using
**Table 1 Classification of 87 genes associated to virulence factors analyzed in this study, based on the genes present in *Helicobacter pylori* (strain ATCC 26695).**

| Virulence factors | Metabolic functions | Related genes |
|---|---|---|
| UREASE | Enzyme; acid resistance; colonization | *ureA* |
| | | *ureB* |
| | | *ureI* |
| | | *ureE* |
| | | *ureF* |
| | | *ureG* |
| | | *ureH/ureD* |
| ADHESINS | Adherence to host cell | *alpA/hopC* |
| | | *alpB/hopB* |
| | | *babA/hopS* |
| | | *babB/hopT* |
| | | *hpaA* |
| | | *hopZ* |
| | | *horB* |
| | | *sabA/hopP* |
| | | *sabB/hopO* |
| LEWIS ANTIGEN | Antigenic mimicry; evasion of the autoimmune response | *futA* |
| | | *futB* |
| | | *futC* |
| IMMUNE MODULATOR (Proinflammatory effect) | Neutrophil activating protein | *napA* |
| | Involved in IL-8 production | *oipA/hopH* |
| FLAGELLA GENES | Motility | *flaA* |
| | | *flaB* |
| | | *flaG* |
| | | *flgA* |
| | | *flgB* |
| | | *flgC* |
| | | *flgD* |
| | | *flgE_1* |
| | | *flgE_2* |
| | | *flgG_1* |
| | | *flgG_2* |
| | | *flgH* |
| | | *flgI* |
| | | *flgK* |
| | | *flgL* |
| | | *flhA* |
| | | *flhB_1* |
| | | *flhB_2* |
| | | *flhF* |
| | | *fliA* |

(Continued)

| Virulence factors | Metabolic functions | Related genes |
|---|---|---|
| | | *fliD* |
| | | *fliE* |
| | | *fliF* |
| | | *fliG* |
| | | *fliH* |
| | | *fliI* |
| | | *fliL* |
| | | *fliM* |
| | | *fliN* |
| | | *fliP* |
| | | *fliQ* |
| | | *fliR* |
| | | *fliS* |
| | | *fliY* |
| CYTOTOXINS | Type IV secretory protein; *CagPAI* (*cag* pathogenicity Island) Secretion system that allows the *cagA* translocation | *cag1* |
| | | *cag2* |
| | | *cag3* |
| | | *cag4* |
| | | *cag5* |
| | | *cagA* |
| | | *cagC* |
| | | *cagD* |
| | | *cagE* |
| | | *cagF* |
| | | *cagG* |
| | | *cagH* |
| | | *cagI* |
| | | *cagL* |
| | | *cagM* |
| | | *cagN* |
| | | *cagP* |
| | | *cagQ* |
| | | *cagS* |
| | | *cagT* |
| | | *cagU* |
| | | *cagV* |
| | | *cagW* |
| | | *cagX* |
| | | *cagY* |
| | | *cagZ* |
| | | *virB11* |
| | Vacuolization of epithelial cells and apoptosis | *vacA* |

| Table 1 (continued) | | |
| --- | --- | --- |
| **Virulence factors** | **Metabolic functions** | **Related genes** |
| PLASTICITY ZONES | Transposons | |
| | Duodenal ulcer promoter | *dupA* |
| | Peptic ulcer promoter | *IceA* |
| | Allelic variants of *IceA* | *iceA1* |
| | | *iceA2* |

SimpleSynteny program (*Veltri, Wight & Crouch, 2016*), and mapping them to genome using Blastn (E-value cutoff = 0.001).

## VF size analysis in base pairs

From of each gene coordinates per strain, an excel matrix was created to obtain the size in base pairs (bp) using Geneious program annotation tool (*Kearse et al., 2012*). From the data, a similarity analysis was performed between each VF per strain, by means of a heat map and the corresponding dendrogram using the programs heatmap.2 (Enhanced Heat Map) (*Khomtchouk, Van Booven & Wahlestedt, 2014*) and the packages "gplots" and "RColorBrewer" in R environment (*Dago et al., 2019*), contrasted with their geographic origin.

## Similarity analysis of virulence factors

Nucleotide identity for each VF by strain was established based on gene similarity percentage by local blastn, global alignment using MUMmer program (*Marçais et al., 2018*) and blast2seq (https://blast.ncbi.nlm.nih.gov/Blast.cgi). Based on an identity matrix, a hierarchical cluster analysis of the structural variables evaluated by strains was performed using the pvclust package (*Suzuki & Shimodaira, 2006*) in R environment.

Each cluster of the hierarchical analysis was supported by $p$-values calculated through multiscale bootstrap resampling. They were compared with two types of $p$-values: AU (Approximately Unbiased) unbiased approximation and BP (Bootstrap Probability).

# RESULTS

## Revision of virulence factors annotation

Our results show that, among the 137 genomes analyzed, 117 are found with gene annotations and 18 genomes, although in Genbank format, do not have any gene annotations. Based on the 87 VFs of *H. pylori* (strain 26695) 9,092 annotated VFs were detected (not including additional copies or genes in unannotated genomes), of which 65.83% (5,986) were confirmed to be annotated while 34.17% (3,106) were identified as annotation errors. From the 3,106 annotation errors, the most frequent are assigned with another gene name with 89.5% (2,779 genes), followed by hypothetical proteins with 6% (187 genes) and genes without annotation but detected in this analysis with 4.5% (140 genes). Blast data and identification by coordinate for annotation errors in each strain are summarized in Table S2.

## Copy number variation of virulence factors in *H. pylori* strains

Our analyses confirm the presence of 11,529 VFs among the 135 genomes analyzed; with an average of 85.4 genes per genome. Of the 87 VFs analyzed, 31 are considered to be completely conserved (the same copy present in all genomes). In this group are all the ureases (six, except *ureA*), three Adhesins (*alpA/hopC, alpB/hopB* and *horB*) and 22 flagella genes (*flaB, flgA, flgB, flgC, flgD, flaG, flgG_2, flgH, flgl, flgK, fliA, fliD, fliE, fliG, fliH, fliI, fliL, fliN, fliQ, fliR, fliS* and *fliY*) are observed among the genomes analyzed. The *flaG* gene, has two identical copies of the gene present in all tested strains. The matrix containing the number of copies per gene in each strain is summarized in the Table S3.

A total of 47 VFs are considered as moderately preserved VFs (most of them with only one copy). In this group are most of the cytotoxins (25 of 28), three Lewis antigen genes (*futA, futb* and *futC*), 12 flagella genes (*flaA, flgE_1, flgE_2, flgG_1, flgL, flhA, flhB_1, flhB_2, flhF, fliF, fliM* and *fliP*), one Urease (*ureA*), four adhesines (*babA/hopS, babB/hopT, hopZ*, and *sabA/hopP*) and two Immune modulator genes (*napa* and *oipA/hopH*). Likewise, nine VFs are considered to be poorly conserved. Two types of variation were found in this group: those with variation at the copy number level (between 0 to three copies per genome) as in the case of adhesine *hpaA* (average of 2.05 copies per genome) and the cytotoxin *virB11* and *vacA* (average of 2.5 and 3.2 copies per genome, respectively). The second group, consisting of VFs of lower frequency (present in only a few genomes) as in the case of the four Plasticity zones genes (*dupA, iceA, iceA1* and *iceA2*) with a average of 0.22 copies per genome; a citotoxine *cag2* with a average of 0.34 copies per genome and a adhesine *sabB/hopO* with a average of 0.32 copies per genome (Table S3).

From the copy number matrix, a dendogram was constructed, showing four well-defined monophyletic groups. These results are represented in Fig. 1. In the monophyletic group called **a**, with 16 genomes (ausabrJ05, K26A1, oki673, B38, oki128, oki154, oki828, 29CaP, BM013A, BM013B, 7C, Aklavik86, Aklavik117, F51, SouthAfrica20 and SouthAfrica7) are included; which do not possess the cytotoxin-related genes of the *cag*PAI (absence of most of the cytitoxins, with the exception of the *virB11* and *vacA* genes which have two to four copies in their genome) (Fig. 1). These *H. pylori* strains have an average of 60 genes found in their genomes (Table S3).

In contrast, the monophyletic group called **b**, with 46 genomes (BM012B, Hp238, 26695-1MET, dRdM1, 26695-dRdM1dM2, 26695-dR, 26695-dRdM2, dRdM2addM2, MKM5, F28, F38, F63, F78, F13, F17, F18, F209, F20, F210, F211, F21, F23, F24, F55, F67, F70, F72, F75, F90, F94, MKF10, MKF3, MKF8, MKM1, MKM6, 26695, 26695-1, 51, F16, F30, F32, F57, OK113, OK310, Rif1, Rif2), is characterized by being the group with the highest number of VF, with an average of 91 copies per genome. In this group of genomes it is observed that the number of copies is very conserved among them. The monophiletic group **c**, includes 8 genomes (2017, Cuz20, Shi112, Shi169, Shi417, Shi470, UM066, XZ274), with an average of 88 copies per genome is characterized by low variability in the copy number of these genes; similar to what was observed in group **d** (L7, DU15, CC33C, PNG84A, G272, HPJP26, 7.13_R1c, 7.13_R3a, 7.13_R2b, 7.13_R1a,

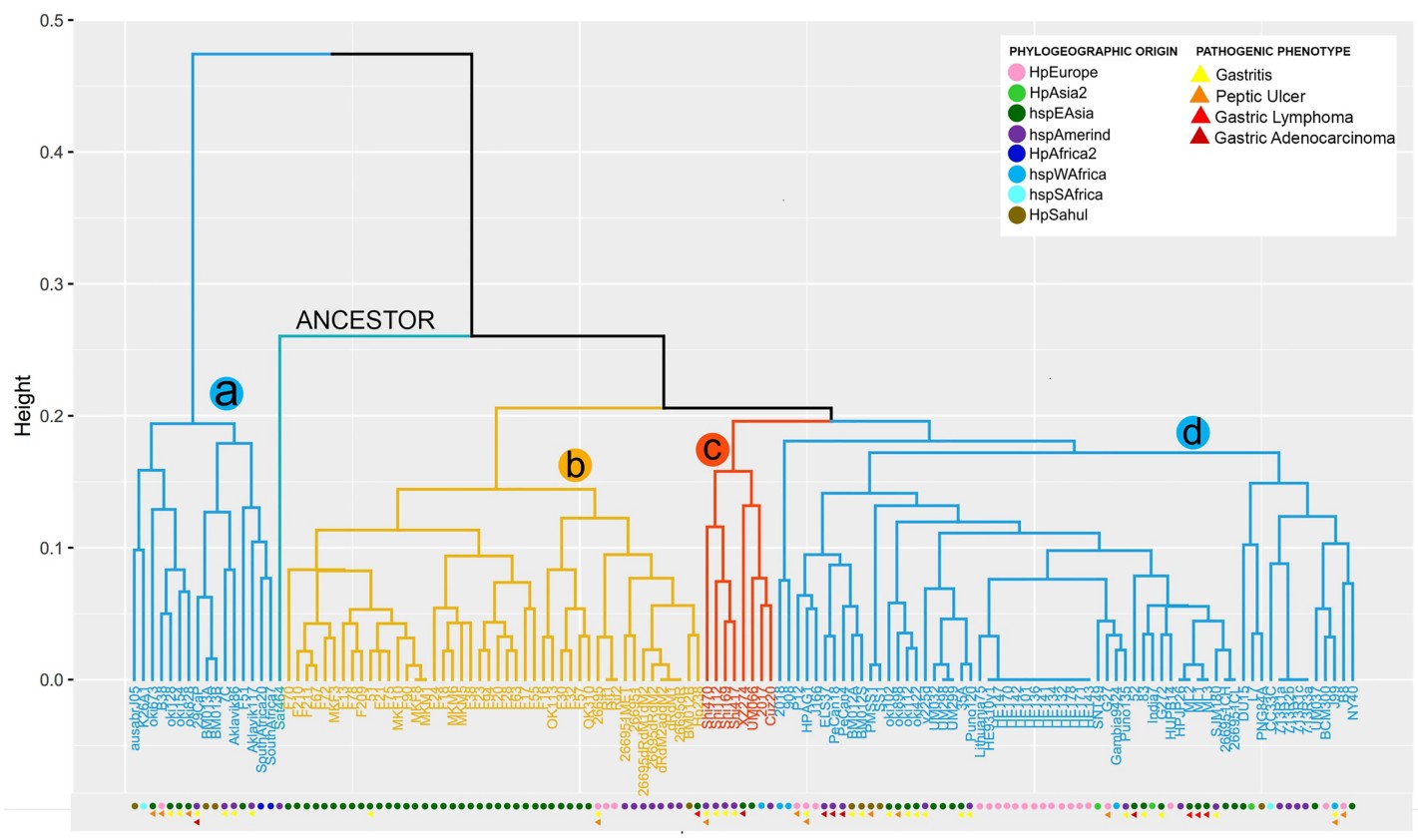

**Figure 1 VF copy number analysis in the 185 *H. pylori* strains.** Colored circles show the phylogeographic origin and colored triangles show the pathogenic phenotype

HE171/09, HE143/09, HE178/09, HE132/09, HE134/09, HE141/09, HE136/09, HE101/09, HE142/09, HE170/09, HE147/09, BCM-300, HE93/10_v1, ML1, ML2, ML3, 2018, 26695-1CH, 26695-1CL, 35A, 52, 83,908, B8, BM012A, BM012S, ELS37, G27, Gambia94/24, HPAG1, HUP-B14, India7, J166, J99, Lithuania75, NY40, P12, PMSS1, PeCan18, PeCan4, Puno120, Puno135, SJM180, SNT49, SS1, UM032, UM037, UM298, UM299, oki102, oki112, oki422, oki898, v225d) (Fig.1, Table S3).

Our analysis shows that 29 strains (26695-1MET, ausabrJ05, K26A1, HPJP26, F28, F38, F51, F55, ML1, ML2, ML3, 26695, 26695-1, 26695-1CH, 26695-1CL, 52, 83, F30, F32, F57, Puno135, Rif1, Rif2, SJM180, SouthAfrica7, UM032, UM298, UM299, and XZ274), have a single copy of *iceA* gene. On the other hand, the presence of two copies of *iceA1* and *iceA2* genes, was observed in 29 strains (BM013A, BM012S, BM013B, 51, ELS37, F13, F16, F21, F67, F70, F72, F75, F78, F90, F209, F210, F211, MKF10, MKF3, MKF8, MKM1, Aklavik86, HPAG1, J166, NY40, OK113, P12 and PeCan4). With the exception of three strains (HPAG1, J166 and P12, which are of European origin). The remaining 77 strains are characterized by the absence of the *iceA* gene (Table S3). By aligning the orthologous regions of strains with only one gene (*iceA*), such as 26695 strain, taken as a reference, *iceA* gene is 519 bp in size compared to the *iceA1* and *iceA2* genes observed

in the European strain B8 with 249 and 390 bp respectively. These results are represented in Fig. S1.

The comparative analysis between copy number in each monophyletic group compared with their phylogeographic origin, indicates that the strains of group **a** and **d** have different origins (Europe, Asia, Africa), while group **b** is dominated by strains of geographic origin of HpEastAsia, covering the Asian and Amerindian population, with some exceptions such as strain 26695 (HpEurope), Rif1 and Rif2 (HpEurope) and BM012B (HpSahul). Likewise, group **c** strains are classified as HpEastAsia (Asian and Amerindian) but with the exception of strain 2017, which belongs to the hspSAfrica subpopulation (Fig. 1).

Despite the reduced information on the pathogenic phenotype of clinical origin of each strain in the databases, group **a** is more likely to develop gastritis and peptic ulcer, compared to group **d** which is more susceptible to develop gastric cancer and gastric lymphoma (MALT) because this group possesses the cytotoxins of the *cag*PAI. However, the limited information obtained made it impossible to relate the number of copies to a pathogenic phenotype in groups **b** and **c** (Fig. 1).

## Virulence factor synteny analyses in *H. pylori* strains

Our positional analyses of the VFs found in the 135 *H. pylori* strains, divided into seven functional groups, showed that most of these genes are present in relatively conserved syntenic blocks. In this sense, Lewis antigen, immune modulator, flagelar genes and cytotoxins genes are highly conserved in both order and orientation in all genomes analyzed. On the other hand, ureases are relatively conserved, presenting a syntenic block of seven compact genes, varying only in strain 2018 (the synteny group is divided into two groups of 3 and 4 genes, with the presence of two genes different from VF) and the absence of *ureA* in strain ausabrJ05 (Fig. 2). Likewise, in the plasticity zone genes, three types of arrangements were observed: (a) strains with a single copy of *iceA* (29 strains), (b) strains with *iceA1* and *iceA2* (29 strains) and (c) strains without any *iceA* gene (77). In this last group, it was observed that strain CC33C presents an additional copy of the *dupA* gene (Fig. 2).

Our analyses indicate that the most rearranged gene family is the adhesins, which include the genes *HpaA, BabA/HopS, BabB/HopT, SabA/HopP, SabB/HopO, alpA/HopC, alpB/HopB, HopZ,* and *HorB*. Figure 2 shows a high variation in the number, order and position of this group of genes in which eight different types of rearrangements are represented. However, within these eight clusters, a total of 66 types of genomic rearrangements in the adhesins were observed that were shared among the 135 *H. pylori* strains (Table S4).

The inversions phylogeny, generated from these adhesins rearrangements, shows a division of three monophyletic groups; **a**, **b** and **c**. The monophyletic group **a** is identified as the most ancestral and whose rearrangement is possessed by strains of the hspEAsia population (Fig. S2). On the other hand, groups **b** and **c** are divided by the inversion in *babA* and *babB* genes, and group **c** is divided into c1 and c2 by the plus/plus position of the s*abA* gene. All ancestral orders (A60 to A117) shown in the phylogeny are summarized

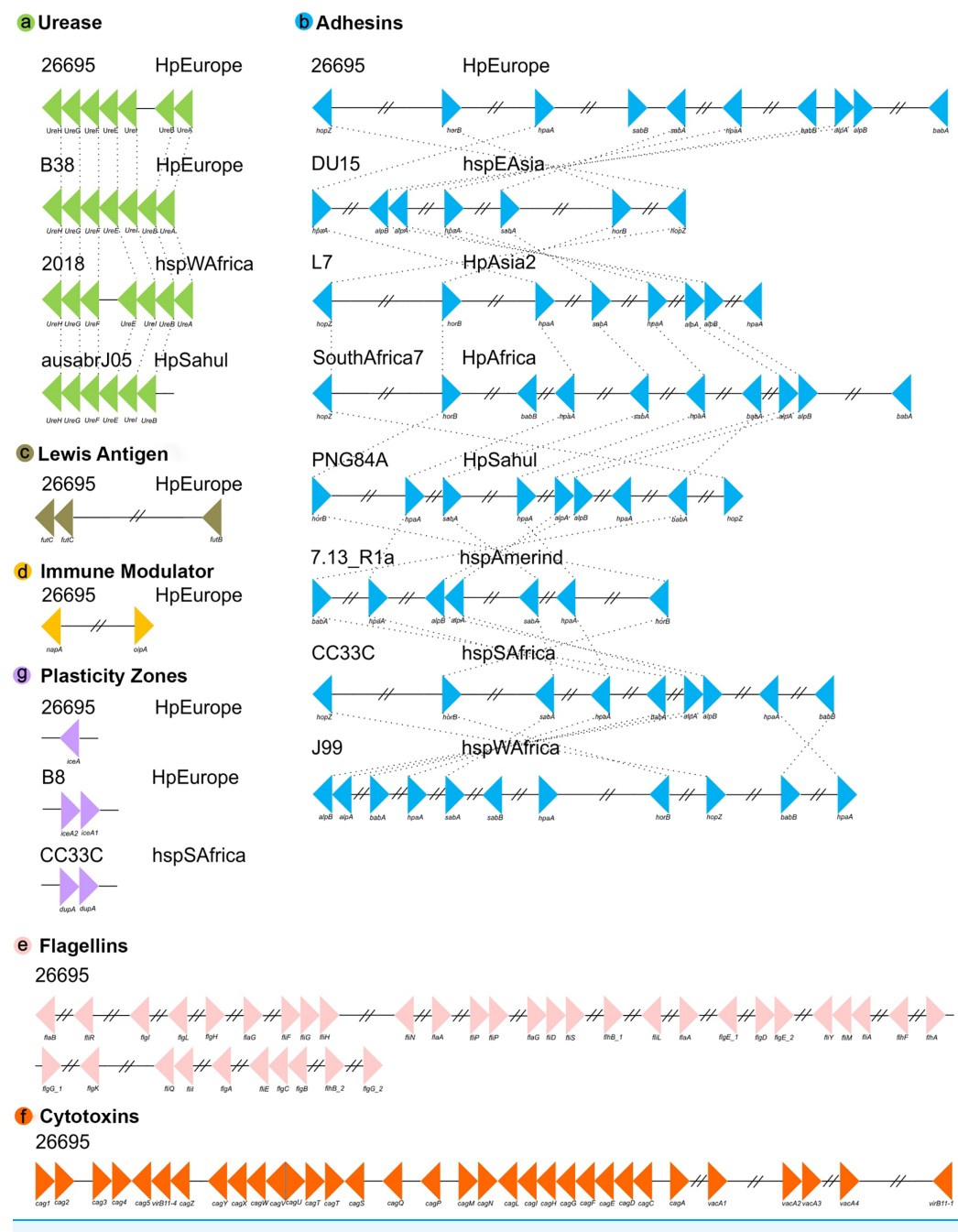

**Figure 2 Synteny block analysis of VF in *H. pylori* strains.** The colored triangles represent the position, order and orientation of each group of VFs: (A) Ureases, (B) adhesines, (C) Lewis antigen, (D) immune modulator, (E) flagellin genes, (F) cytotoxins and (G) plasticity zones in the *H. pylori* strains analyzed. The order is represented by a representative strain with its respective phylogeographic origin

in Table S5. However, regarding the adhesin inversions phylogeny (Fig. S2), there is no evidence of a clear phylogeographic relationship, nor a relationship with the pathogenic phenotype among the 66 rearrangements analyzed.

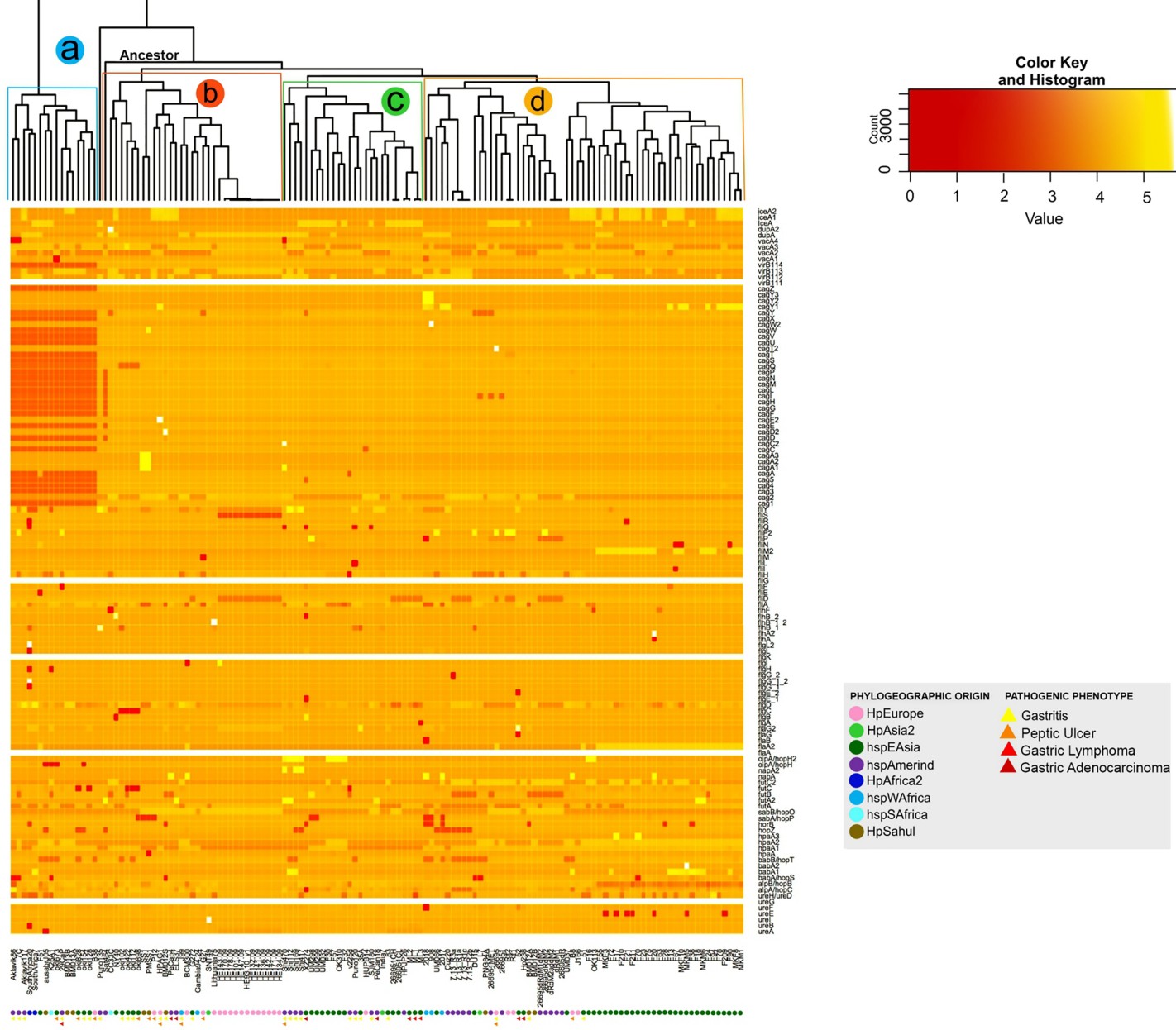

**Figure 3 Histogram of intraspecific difference in bp size between VF of *H. pylori*.** Variation in each gene per strain is represented by conserved (light colors) and less conserved (dark red) genes. Colored circles show the phylogeographic origin and colored triangles show the pathogenic phenotype.

## Intraespecific difference in size between virulence factors of *H. pylori*

The analysis of the size in bp of each of the VFs in the 135 *H. pylori* strains, show a significant intraespecific variation in size in base pair can be observed in most of the VFs (from the coordinates of each gene in the genome, see Table S2). As a result, the dendogram shows four distinct monophyletic groups (**a**, **b**, **c** and **d**) (Fig. 3). In this

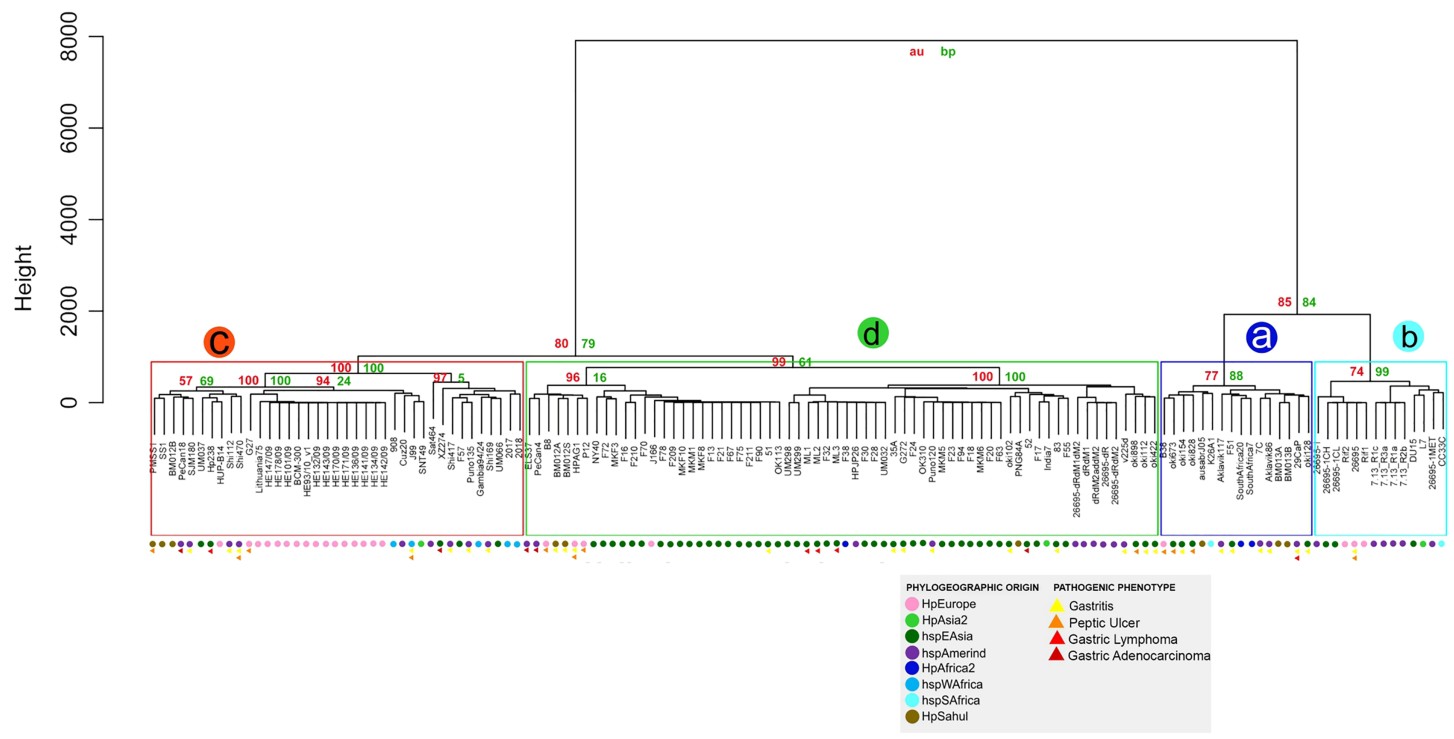

**Figure 4 Hierarchical clustering dendrogram from the similarity of 87 VF in *H. pylori* strains.** Colored circles show the phylogeographic origin and colored triangles show the pathogenic phenotype.

analysis, the group **a** (Aklavik86, 7C, Aklavik117, SouthAfrica20, SouthAfrica7, F51, ausabrJ05, K26A1, oki673, 29CaP, BM013A, BM013B, oki828, oki154, oki128 and B38) are the strains that do not present the genes belonging to the *cag*PAI (red color in Fig. 3); while a total conservation in size of five genes (*ureG, flaA, flgK, fliG* and *virB11-1*; white color) is observed for all 135 strains analyzed. Despite the presence of the pathogenicity island genes, the monophyletic groups **b**, **c** and **d** (Fig. 3) are similar to those observed in the copy number dendrogram (see Fig. 1).

On the other hand, strains of the monophyletic group **a** are associated with the development of gastritis and peptic ulcer disease. However, groups **b**, **c** and **d** do not present a visible pathogenic pattern. Likewise, the monophyletic group **d** is associated with the East Asian geographic lineage. The other groups of strains show no association with their phylogeographic origin (Fig. 3).

## Similarity of virulence factors in *H. pylori*

The similarity analysis shows results congruent with the previous analysis, with four monophyletic groups composed of the same strains, where group **a** also groups the strains that do not have the *cag*PAI. Groups a and b present a lower percentage of identity and *p*-value of 70% to 90% compared to groups **c** and **d**, where the strains are grouped with a high similarity of 90% to 100% demonstrating the homology of these genes with respect to the reference genes (Table S6 and Fig. 4).

## DISCUSSION

### Revision of virulence factors annotation in *H. pylori* strains

Our analysis shows a significant percentage (34.17%) of genes with annotation errors in the *H. pylori* strains analyzed; most of them (89.5%) due to misidentification of the gene by gene ontology. Previous studies have reported that the potential errors of the first three published genomes of *Haemophilus influenzae*, *Mycoplasma genitalium* and *Methanococcus jannaschii* showed that, depending on the type of function, the expected rate of errors varies from less than 5% to more than 40% (*Devos & Valencia, 2001*). Further studies estimates of error rates in curated sequence annotations stayed at the same level of 28–30% (*Jones, Brown & Baumann, 2007*); similar to detected in *Campylobacter jejuni* genome (*Gundogdu et al., 2007*).

Current sequencing methods produce hundreds of bacterial genomes deposited in databases such as the NCBI database, which are annotated in an automated process. Genome annotation has become a critical element for us to understand genomic biology, especially the genomes of pathogenic microorganisms (*Dong et al., 2021*). However, genome annotation is a crucial step for the extraction of useful information from genomes so that errors in genic annotation are relatively frequent because of the lack of sufficient data, and these errors might propagate into other genomes (*Zhang, Li & Zhou, 2014*). According to *Denton et al. (2014)* several genomes (particularly prokaryotes) are usually first-drafts, with a lot of missing data, many gaps, and lot of errors in the published sequences mainly by incomplete genome assemblies. *Salzberg (2019)* proposes that the main challenges in genome annotation could be due to automated annotation of large, fragmented "draft" genomes, and contamination in draft assemblies leading to errors in annotation that tend to propagate across species. As genome sequencing continues to accelerate and as erroneous annotations are sometimes used as the basis for further genome annotations, resulting in what has been called a "percolation of errors", effect common in mammalian mitochondrial genome (*Prada & Boore, 2019*); so that an inaccurate genome annotation may influence subsequent studies. Since old errors may propagate to the newly sequenced genomes, *Poptsova & Gogarten (2010)* emphasize that the problem of continuously updating popular public databases is an urgent and unresolved one. In this regard, protein-coding gene detection in prokaryotic genomes is considered a much simpler problem than in intron-containing eukaryotic genomes, the number of missing genes in the annotation of prokaryotic genomes is worryingly high (*Warren et al., 2010*).

One of the effects of the annotation errors reported in this work is the false negatives of paralogous genes in certain strains of *H. pylori*; with serious implications in associating the genotype with the pathogenic phenotype of these bacteria. Due to the high percentage of errors in the annotation of clinically important genes such as virulence factors in pathogenic bacteria such as *H. pylori*, it is necessary a semi-automated procedure analysis such as ours, which proposes a procedure for the reannotation of these strains.

### Genetic variability of virulence factors in *H. pylori* strains

Our results show three main groups of VF: highly conserved, moderately conserved and poorly conserved among the *H. pylori* strains analyzed.

### Highly conserved virulence factors

A significant number of completely conserved genes among the strains analyzed, which could be interpreted as "basal VF in *H. pylori*". Within these conserved genes are the most of the ureases (except *ureA*); which plays an essential role in stomach colonization by metabolizing urea into ammonia in order neutralize stomach acid needed to permit survival in the gastric compartment (*Mannion, Shen & Fox, 2018*). Although our analyses show the absence of the *ureA* gene in ausabrJ05 (HpSahul) strain, there is no information that proves the greater or lesser pathogenicity due to the absence of this gene. In addition to being conserved among strains (one copy per gene per genome), urease genes are conserved in size, identity and synteny. Earlier studies have shown that negative urease mutant strains, built by inserting resistant antibiotic cassettes in the *ureA*, *ureB* and *ureG* genes, lost urease activity (*Ferrero et al., 1992*); lacked the ability to colonize the mammal stomach, demonstrates urease is essential for chronic infection (*Debowski et al., 2017*; *Tsuda et al., 1994*). This is evidence of their fundamental role in the process of host-host interaction.

Likewise, our results show that most of the flagellar genes (22 of 34) are conserved in *H. pylori* strains. The flagellum consists of three basic structures referred to as the basal structure, the hook, and the filament; organelle that are involved not only in motility and chemotaxis and participate in many additional processes including adhesion, biofilm formation, virulence factor secretion, and modulation of the immune system of eukaryotic cells, contributing to bacterial pathogenicity in *H. pylori* (*Duan et al., 2013*; *Ramos, Rumbo & Sirard, 2004*). Therefore, the presence of these 22 conserved flagellar genes in all strains of *H. pylori* could be associated with these important cellular functions; which are part of ancient core set of flagellar structural genes that were present in the common ancestor to all Bacteria (*Liu & Ochman, 2007*). Similar to ureases, these flagellin genes are conserved among strains in copy number (one copy per gene per genome), in size, identity and synteny.

The presence of three adhesins (*alpA/hopC, alpB/hopB and horB*) conserved in copy number in all strains, would indicate that these genes would be strongly implicated in the adherence of *H. pylori* to the mucus layer of the gastric epithelium (*Burucoa & Axon, 2017*; *Javed, Skoog & Solnick, 2019*; *Peleteiro et al., 2014*; *Šterbenc et al., 2019*). Similarly, it has been demonstrated that these genes plays an important role in the initial colonization and persistence of the bacteria in the human stomach during decades or for the entire lifetime (*Oleastro & Ménard, 2013*).

### Moderately conserved virulence factors

On the other hand, our analysis shows that 12 flagella genes (*flaA, flgE_1, flgE_2, flgG_1, flgL, flhA, flhB_1, flhB_2, flhF, fliF, fliM and fliP*) are considered moderately conserved; due to the gain or loss of a gene in a given genome. For example, 52 and XZ274 strains,

have a deletion of one copy of *flgE* gene. *flgE* is the main protein of the flagellar hook, and strains lacking the *flgE* gene expectedly showed no motility (*O'Toole, Kostrzynska & Trust, 1994*). However, all *H. pylori* strains with excision of these two, present two copies of the gene (*flgE_1* and *flgE_2*), so it is presumed that in these two strains with the deletion of one of the copies, the mobility could be reduced but not totally. Nevertheless, according to our analysis, these two strains are associated with the formation of Gastric adenocarcinoma, so it would not be clear the relationship between the absence of this gene with its pathogenic phenotype. Similarly, DU15, 35A, CC33C and 29CaP strains show deletion of the *flhB, fliF* and *flhF* genes, respectively. Based on previous studies, *flhB* and *fliF* mutant strains did not produce any flagella and were non-motile, which would imply a serious reduction in the colonizing ability of these strains (*Allan et al., 2000*; *Gu, 2017*; *Tsang & Hoover, 2015*).

On the other hand, duplications in six flagella genes were observed in 30 *H. pylori* strains; strains possessing certain copy number characteristics as they are mainly grouped in the monophyletic group b. Additionally, some of these duplications are associated with a phylogeographic origin, as in the case of an additional copy of the *fliM* gene present in strains 908, F20, F211, F21, F23, F24, F55, F67, F70, F72, F75, F90, F94, MKF10, MKF3, MKF8, MKM1 and MKM6, which, except for 908 (of African origin), are of Asian origin. However, the presence of an additional copy of the *flaA, flgG, flgL, flhA, fliM* and *fliP* genes is associated with a small number of strains without a clear phylogeographic origin or pathogenic phenotype. Although the dynamic variation in gene dosage plays a vital role in both adaptation to changing conditions and the generation of novel genes in pathogenic bacteria (*Andersson & Hughes, 2009*; *Elliott, Cuff & Neidle, 2013*), it is not clear how the acquisition of a new copy of a certain flagellar gene in a strain can be associated with the development of a certain pathology such as cancer.

Our results show that the Lewis antigens such as *futA* and *futB* genes are found in 75 and 83% of the strains, respectively; similar to those previously observed (*Qumar et al., 2021*). By contrast, the *futC* gene is not only present in 129 of the 135 strains but it also presents a second copy in 38 strains. Moreover, it is relatively preserved in copy number; the Lewis antigens are also moderately conserved among strains in copy number, in size and identity, but highly conserved in position between the strains.

## Virulence factors poorly conserved

Our results shown that the adhesins *babA/hopS* gene shows a copy number variation, with an absence of this gene in the six strains (7C, L7, DU15, K26A1, F70 and Aklavik86), or an additional copy in the 11 strains (MKM5, F38, F13, F18, F90. MKF10, MKM6, B8, Shi112, Shi169 and SouthAfrica7). According to previous studies, the *H. pylori babA* adhesin facilitates the binding of *H. pylori* to the fucosylated Lewis b histo-blood group antigen which is present on the surface of gastric epithelial cells, thus facilitating colonization and determining bacterial density (*Guruge et al., 1998*; *Šterbenc et al., 2019*). These results are consistent with previous results showing that some *H. pylori* strains have a single copy of the gene and others have two of the *babA* gene (designated *babA1* and *babA2*) in which heterogeneity among *H. pylori* strains in expressing the *babA* protein may

 

be a factor in the variation of clinical outcomes among *H. pylori*-infected human (*Hennig et al., 2004*; *Šterbenc et al., 2019*).

Likewise, deletion in *babB/hopT*, *hopZ*, and *sabA/hopP* genes were observed in 21, 13 and 9 strains, respectively (See Table S2). Themselves, the results show a very low number of *sabB/hopO* genes (43) in the strains analyzed. According to *de Jonge et al. (2004)*; the off-status of *sabB* was found to be associated with duodenal ulcer disease, and thus represents a putative marker for disease outcome. However, according to our analysis, the presence or absence of the *baba*, *babB/hopT*, *hopZ*, *sabA/hopP* and *sabB/hopO* genes is not clearly related to a phylogeographic origin or pathogenic phenotype. Nevertheless, recent studies show that, patients infected with strains carrying *iceA1*, *sabA* "on" and *hopZ* "off" had 10-fold higher odds (OR = 10.3, 95% CI [1.2–86.0]) of developing MALT lymphoma than age-matched patients with gastritis (*Šterbenc et al., 2019*).

Gene copy number variation in bacteria is probably severely underreported, and there are very few reports on the regional distribution of the phenomenon (*Brynildsrud et al., 2016*); which demonstrates that analysis of copy number variation in a given combination of deletions and duplications of one or more genes of different metabolic pathways are key to pathogenic behavior in *H. pylori*. Furthermore, the ability to examine genomic change in pathogenic bacteria provides insight into virulence and genetic adaptation to host environments (*Bryant, Chewapreecha & Bentley, 2012*). Despite the fact that many studies show that the phenomenon of gene duplication in bacteria is frequent, the complexity of host–pathogen interactions can obscure the role of gene expansion in adaptive responses (*Elliott, Cuff & Neidle, 2013*).

Although in size and identity they do not present great variations, the adhesins present great variation at the genic order level. Our analyses show a high level of reorganizations in these adhesins, such as inversions along the genomes analyzed; grouping certain strains in a phylogeographic and pathogenic sense (strains of the hspEAsia population, having inversion in the *hpaA* and *babA* genes with a higher probability of developing gastritis and peptic ulcer). The inversions of one or more genes might be fixed in species due to direct mutational effects associated with inversion breakpoints located near or inside genes, which might affect their function and/or expression profile; or known as "position effect" hypothesis (*Sperlich, 1986*). According to the position effect hypothesis, these features might have implications for gene expression patterns and would place the encoding region in a different regulatory context (*Frischer, Hagen & Garber, 1986*). Recently, it has been shown that gene inversion potentiates bacterial evolvability and virulence in 12 pathogenic bacterial species, including *Campylobacter jejuni* (*Merrikh & Merrikh, 2018*). Therefore, our analysis may be the first evidence of a possible position effect between *H. pylori* strains.

According to our analysis, the presence and absence of genes belonging to the *cag*PAI pathogenicity island is a clear dichotomous feature in the molecular characterization of *H. pylori* strains. Although a relationship between the *cag*PAI-negative and the development of a phenotype such as gastritis or peptic ulcer disease has been observed. Likewise, of the 16 *cag*PAI-negative strains, six are hspEAsian, six are African, three are hspAmerindian and one of HpEuropean origin. Nevertheless, our analyses are not

conclusive with a specific pathogenic phenotype or phylogeographic origin. However, different studies have determined that the integrity of *cag*PAI seems to have an important role in the progress of the gastroduodenal disorders, so that intact *cag*PAI could be seen in *H. pylori* strains from countries with higher rate of gastric cancer (*Lai et al., 2013*; *Parsonnet et al., 1997*). Several studies have investigated the association of *H. pylori* PAI and gastroduodenal diseases (*Khatoon et al., 2017*; *Lai et al., 2013*). *Hanafiah et al. (2020)* found an association of *cag*PAI intactness with histopathological scores of the gastric mucosa. In this work, *H. pylori* harbouring partial *cag*PAI were associated with higher density of *H. pylori* and neutrophil activity, whereas *H. pylori* with deleted *cag*PAI caused increased in inflammatory score.

The presence of the *cag*PAI region is almost universal in *H. pylori* hpEastAsia and hpAfrica1 populations, intermediate presence in hpEurope, and complete absence in hpAfrica2 (*Olbermann et al., 2010*). A recent study in multiracial Malaysian population show that of 96.6% (*n* = 85) of *H. pylori* isolates were *cag*PAI-positive with 22.4% (19/85) having an intact *cag*PAI, whereas 77.6% (66/85) had a partial/rearranged *cag*PAI (*Hanafiah et al., 2020*). Based on these results, the authors propose that the variation in the *cag*PAI positivity in different population of *H. pylori* isolates might be related to different geographical origin of *H. pylori* subpopulations.

Another group of genes highly variable in copy number are plasticity zones. Different studies demonstrated that certain genes in this region may play important roles in the pathogenesis of *H. pylori*-associated diseases. Plasticity zone cluster is a virulence factor that may be important for the colonization of *H. pylori* and to the development of severe outcomes of the infection with *cagA*-positive strains (*Ganguly et al., 2016*). According to our analysis, the *dupA* gene is present in 28.8% (39/135) of the strains analyzed, where 38.5% (15/39) of the strains are of HpEuropean origin, 23.1% (9/39) are hspEAsia, 20.5% (8/39) are hspWAfrica and 17.9% (7/39) are hspAmerica and 17.9% (7/39) are hspAmerica. Previous reports indicate that infections with *dupA*-positive strains increased the risk for duodenal ulcer, but they were protective against gastric atrophy, intestinal metaplasia and gastric cancer (*Lu et al., 2005*; *Shiota, Suzuki & Yamaoka, 2013*). Our results are consistent with those presented by *Alam et al. (2020)* where they indicate that the prevalence of *dupA*-positive isolates is around 40% in Asian, North African and South American populations; associated with duodenal ulcer.

Likewise, the majority of the strains (77) did not contain the *ice*A gene, 29 strains have a single copy of *iceA* gene, while 29 strains have two copies of *iceA1* and *iceA2* genes. When analyzing these strains with only one *iceA* gene, there is no relationship with the clusters observed in the copy number dendrogram or by geographic origin. However, oun results shown that the presence of *iceA1* and *iceA2* copies is related to an Asian/Amerindian geographic origin.

The epithelium antigen gene (*iceA*) was identified in the *H. pylori* isolated from peptic ulcer disease and gastritis patients; with at least two alleles of *iceA*, *iceA1*, and *iceA2* (*Yakoob et al., 2015*). Several studies suggest an association of the *iceA1* variant and peptic ulcer disease (*Amjad et al., 2010*). On the other hand, iceA2 has no homology to known genes, and the function of the *iceA2* product remains vague in spite of the fact that this

allele is associated with asymptomatic gastritis and nonulcer dyspepsia (*Abu-Taleb et al., 2018*; *Amjad et al., 2010*). Based on our analysis, it is proposed to a fission event of the *iceA* gene, which generated the two alleles *iceA1* and *iceA*. In this case, a possible neofunctionalization of the *iceA* gene, where one copy of the duplicated gene maintains the original function and the other acquires a new function different from the original but evolutionarily more advantageous (*Qian & Zhang, 2014*). However, the absence of a pathogenic phenotype in most of the strains analyzed limits the association between genetic variability and pathogenic phenotype. Therefore, it is proposed that all the strains sequenced have detailed information on their pathogenic phenotype for future research. Also, it would be very important to carry out a study of genetic variability of antibiotic resistance genes, the presence or absence of plasmids, and their relationship with the pathogenic phenotype of each strain.

## CONCLUSIONS

The analysis of the *H. pylori* genomes available in the database shows a significant rate of gene annotation errors. We judge that the application of simple bioinformatic tools in the verification of gene annotation, particularly for virulence factor genes, would be a very useful enhancement for the curation of phatogenic bacterial genome sequences submitted to GenBank. Likewise, our results indicate that rearrangements as duplications and deletions of one or more genes represent an important change in the *H. pylori* genome. Our results show that there are a large number of basal or highly conserved VFs among the strains, and another group of VFs that would be responsible for the genetic variability among *H. pylori*. The finding of this study enhance our understanding of *H. pylori* genome and its association to their geographic origin and pathogenicity.

### Funding

This work was supported by Universidad del Tolima, Colombia. The Oficina de Investigaciones y Desarrollo Científico de la Universidad del Tolima for postdoctoral fellowships (4/2019) supported Carlos F Prada Quiroga. The funders had no role in study design, data collection and analysis, decision to publish, or preparation of the manuscript.

### Competing Interests

The authors declare that they have no competing interests.

### Author Contributions

- Aura M. Rodriguez conceived and designed the experiments, performed the experiments, analyzed the data, prepared figures and/or tables, and approved the final draft.
- Daniel A. Urrea conceived and designed the experiments, analyzed the data, prepared figures and/or tables, authored or reviewed drafts of the paper, and approved the final draft.

- Carlos F. Prada conceived and designed the experiments, performed the experiments, analyzed the data, prepared figures and/or tables, authored or reviewed drafts of the paper, conceived and designed the experiments, and approved the final draft.

## Data Availability

The raw data are available in a Supplemental File.

## Supplemental Information

Supplemental information for this article can be found online at http://dx.doi.org/10.7717/peerj.12272#supplemental-information.

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
