# Peer review of "Helicobacter pylori virulence factors: relationship between genetic variability and phylogeographic origin"

_PeerJ, doi:10.7717/peerj.12272_

## Round 0.1 · original submission · Major Revisions

Reviewers have raised some serious concerns and shortcomings in the study. MAJOR revision is suggested and requires substantial and thorough revision to improve the quality of the manuscript. Therefore, authors are requested to revise their manuscript in light of reviewers' comments. Please justify and discuss all questions raised by the reviewers and resubmit the revision accordingly.

Reviewer 1 ·

Basic reporting

• Comment 1: Plasticity region is flexible genes. Four genes of plasticity region could not determine that whole region.
• Comment 2: Lines 255-260, please include the analysis and p value.
• Comment 3: From copy number matrix, does it keep groups without cagPAI genes? Because this groups look like more defined due to cagPAI genes existence not other genes.

Experimental design

• Comment 4: I am not sure that positional analysis and rearrangements of adhesin can correlate virulence of H. pylori. Because adhesin or OMPs activity is under functional condition and some of them have on or off type (Y.Yamaoka et al., Gut, 2006). What was your hypothesis before your analysis for this part?
• Comment 5: Please write about possibility of monophyletic group A (EAsian population) can be ancestral strains. Also, at lines 286-288, if there was no relationship between phylogeographic and 66 rearrangements, previous sentence (lines 281-282) was no meaning.
• Comment 6: Please mention about study limitation and further direction in the discussion.

Validity of the findings

• Comment 7: Please correct all H. pilory to H. pylori
• Comment 8: Please describe the cut-offs of identity and coverage percentages using VFs detection in method section.
• Comment 9: Conclusion should be clear and summarized the key supporting ideas with less hypothetical.

Additional comments

The manuscript named “Helicobacter pylori virulence factors: relationship between genetic variability and phylogeographic” tried to identify correlations between virulence factors, its copy numbers and phylogeographic origin.

Generally, the research question is good, and explanation of result were not insufficient. There are several points that may help the paper become more impactful as shown in each section.

Reviewer 2 ·

Basic reporting

Dear Editor
The review of the manuscript with title: Helicobacter pylori virulence factors: relationship
between genetic variability and phylogeographic origin has been finished and i think in addition of the virulence factors if authors wrote about the antibiotic resistance genes it could be very interesting and informative.

Experimental design

About the design of the study its very good.

Validity of the findings

The finding is very interesting, but about the validity as authors indicated this study is based on bioinformatic and may be in practice the data have some controversy and need to done in practice.

Additional comments

As i wrote in above, if the authors mention the antibiotic resistance genes, i think the study could be very interesting and informative.
Best Regards

Reviewer 3 ·

Basic reporting

This bioinformatic study has analyzed 135 strains of H. pylori genome to determine genome variation and its relation to VF with respect to their geographic distribution. In general, the manuscript is written in clear English but same sentences are unnecessarily long and flue that require revision. Sufficient amount of data have been gathered and evaluated in an appropriate tools and methods. All findings has been appropriately presented in general and discussed in the light of sufficient and related literature. Overall, this study has been achieved successfully to reach its aims. In my opinion, finding of this study enhance our understanding of H. pylori genome and its association to their pathogenicity.
Besides, some comments, suggestions and corrections has been done on the attached manuscript that required to be fulfilled.

Experimental design

Genomes were analyzed by appropriate bioinformatic tools and methods. This is an original bioinformatic research within the scope of the PeerJ journal. The question is well defined in the introduction section, clearly proves the gaps in our knowledge and clarify the importance of the study.
Sufficient amount of data were analyzed with well defined advanced bioinformatic tools and well evaluated in accordance with the aim of the study.
On the other hand, some comments and corrections need to be made on the attached manuscript.

Validity of the findings

No doubt that Findings of this study would enhance our knowledge on the genome variation and its relation to the pathogenicity of H. pylori. It is clear that data are robust but there are some ambiguities related to the genomes stored in the Genbank that is taken into consideration and well discussed in the discussion. Conclusion are made reasonably but require some amendments as stated on the attachment.

Annotated reviews are not available for download in order to protect the identity of reviewers who chose to remain anonymous.

---

## Round 0.2 · Minor Revisions

The manuscript is significantly improved by the authors. However, there are still some minor concerns raised by the reviewer. Please address these concerns and resubmit accordingly.

Reviewer 1 ·

Basic reporting

The revised version is well-written.

Experimental design

The revised version is well-written.

Validity of the findings

The revised version is well-written.

Additional comments

none

Reviewer 3 ·

Basic reporting

The manuscript has generally been revised and corrected based on my comments. (1) On the other hand, there are some typos like lewis (line 113), citotoxins (line 263). (2) The citation of references in the text and the style of the text in the references were not written according to PeerJ's instructions. Please correct all references throughout the text. (3) I would recommend having the English checked by a native speaker. (4) Some corrections have not been made in the so-called revised PDF copy.

Experimental design

Done

Validity of the findings

Done

---

## Round 0.3 · accepted · Accept

The manuscript is significantly improved by the authors and now can be accepted in its current form.